# Downscaling of SMAP Soil Moisture in the Lower Mekong River Basin

**Chelsea Dandridge \*, Bin Fang and Venkat Lakshmi**

Department of Engineering Systems and Environment, University of Virginia, Charlottesville, VA 22904, USA; bf3fh@virginia.edu (B.F.); vlakshmi@virginia.edu (V.L.)
**\*** Correspondence: cld9mt@virginia.edu; Tel.: +1-434-547-2207

**Abstract:** In large river basins where in situ data were limited or absent, satellite-based soil moisture estimates can be used to supplement ground measurements for land and water resource management solutions. Consistent soil moisture estimation can aid in monitoring droughts, forecasting floods, monitoring crop productivity, and assisting weather forecasting. Satellite-based soil moisture estimates are readily available at the global scale but are provided at spatial scales that are relatively coarse for many hydrological modeling and decision-making purposes. Soil moisture data are obtained from NASA's soil moisture active passive (SMAP) mission radiometer as an interpolated product at 9 km gridded resolution. This study implements a soil moisture downscaling algorithm that was developed based on the relationship between daily temperature change and average soil moisture under varying vegetation conditions. It applies a look-up table using global land data assimilation system (GLDAS) soil moisture and surface temperature data, and advanced very high resolution radiometer (AVHRR) and moderate resolution imaging spectroradiometer (MODIS) normalized difference vegetation index (NDVI) and land surface temperature (LST). MODIS LST and NDVI are used to obtain downscaled soil moisture estimates. These estimates are then used to enhance the spatial resolution of soil moisture estimates from SMAP 9 km to 1 km. Soil moisture estimates at 1 km resolution are able to provide detailed information on the spatial distribution and pattern over the regions being analyzed. Higher resolution soil moisture data are needed for practical applications and modelling in large watersheds with limited in situ data, like in the Lower Mekong River Basin (LMB) in Southeast Asia. The 1 km soil moisture estimates can be applied directly to improve flood prediction and assessment as well as drought monitoring and agricultural productivity predictions for large river basins.

**Keywords:** SMAP; passive microwave soil moisture; soil moisture downscaling

## 1. Introduction

Estimating the water balance in large watersheds is of great interest for water resource management and soil moisture is a key variable in this estimation as it effects evaporation, infiltration, and runoff [1]. Soil moisture acts as a link between energy and water fluxes at Earth's surface-atmosphere interface, and knowledge of soil moisture variation is the key to understanding the hydrological cycle [2]. Soil moisture is the primary source of water for agriculture and directly influences crop growth and food production [3]. Even though it only accounts for a small portion of global freshwater, it is still an important factor in global hydrologic cycles [3]. This seemingly small layer (top few centimeters) controls the regulation and distribution of precipitation between runoff and water storage [4]. Soil moisture observations over large areas are increasingly necessary for a range of applications such as meteorology, hydrology, water resource management, and climatology [5].

Remote sensing has provided valuable data sets for understanding land surface hydrological and meteorological processes [6–9].

Obtaining soil moisture measurements can be achieved using a variety of remote sensing instruments or ground-based systems. Satellite-based radars can measure soil moisture at high resolution but are limited in spatial coverage and temporal frequency. Satellite data products can produce global soil moisture estimates but are usually too coarse for practical use in modelling and decision-making [10]. High resolution soil moisture estimates can be applied directly to improve flood prediction and assessment as well as drought monitoring, agricultural productivity prediction, and irrigation management [11–14]. With improved prediction of extreme events, we can also better prepare for their effects on the natural environment and future climate change [2]. NASA's soil moisture active passive (SMAP) will help determine whether there will be more or less water, regionally, in the future compared to today [15,16]. Monitoring these changes in future water resources is a very important aspect of climate change as this will affect the future water supply and food production in areas like the Lower Mekong Basin [17–19]. High resolution soil moisture can aid in crop yield forecasting as well as by providing earlier monitoring of droughts and better understanding of hydrologic processes [4].

This research uses global soil moisture data derived from the L-band radiometer aboard NASA's SMAP observatory [20]. However, satellite microwave radiometers are much coarser than active microwave and optical systems [6]. This coarseness reduces satellite applicability in large watershed models and for regional flood prediction [21]. This study aims to downscale SMAP soil moisture estimates, from gridded 9 km resolution to 1 km resolution, in the Lower Mekong Basin (LMB). This will be done using the regression relationship between daily temperature changes and daily soil moisture under different vegetation conditions with the algorithm developed by Fang et al., 2013. Soil moisture estimates with high spatial resolution can be very useful for watershed scale hydrological modeling due to the fact that soil moisture estimates can be used to constrain errors during extensive wetting and dry downs [21]. The downscaling algorithm and methodology implemented in this research were developed in a previous study by Fang et al., 2018. This algorithm has been applied to the Black Bear-Red Rock watershed in Oklahoma and validated with in situ soil moisture from the ISMN (International Soil Moisture Network). Regions with low elevation are vulnerable to flooding and other water-resource related problems. With these problems, it is important to increase the capacity of flood and drought monitoring. Here we apply this validated algorithm to the Lower Mekong Basin, an area with no functioning in situ soil moisture network. With higher resolution soil moisture, this region would have greater modelling capabilities and the ability to make better decisions concerning water resource management. This algorithm can be applied to other watersheds worldwide, with little absent from the in situ soil moisture systems.

The Mekong River in Southeast Asia provides food, water, and energy resources to the countries of China, Laos, Myanmar, Thailand, Cambodia, and Vietnam [2]. It is the 12th longest river in the world, extending over 4300 km [22]. The basin can be divided into two major catchments also known as the upper and lower river basins. The upper basin is mostly mountainous, rising in the Tibetan Plateau (Figure 1). The Lower Mekong Basin (LMB) is subject to high levels of flooding due to the combination of low-lying terrain and seasonal precipitation cycles [22]. The LMB is home to the rice paddy fields of Vietnam, which would benefit greatly from consistent soil moisture data. Unfortunately, the LMB does not have a consistent in situ soil moisture measuring system, which makes satellite-derived soil moisture estimates appealing for application in watershed-scale hydrological modelling in this region. The lack of ground measurements for soil moisture also complicated the validity of remotely-sensed estimates of the LMB [2].

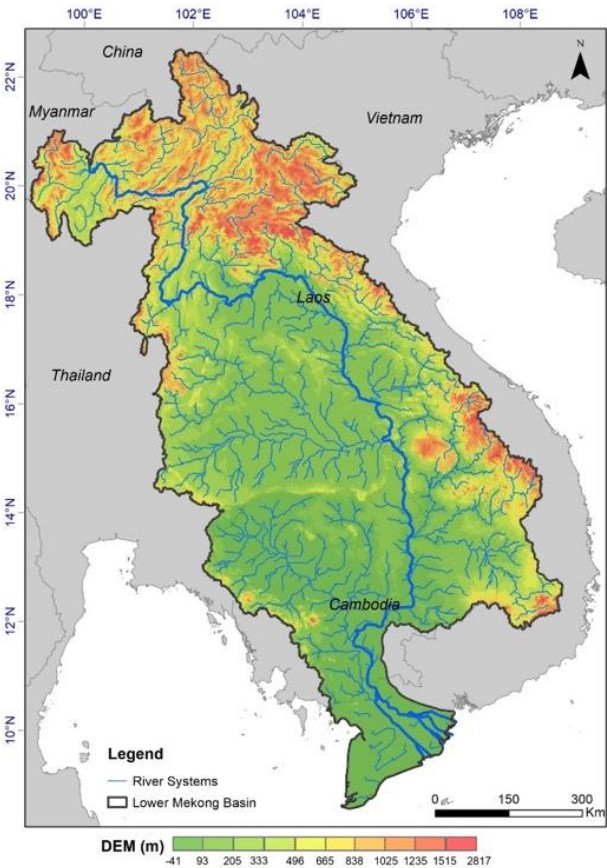

**Figure 1.** Topography and river networks in Lower Mekong River Basin (LMB).

## 2. Data

### 2.1. SMAP Data

Developed by NASA, the soil moisture active passive (SMAP) observatory was designed to distinguish between frozen and thawed land surfaces [14]. This mission was launched in January 2015 with the goal of combining radar and radiometer at L-band frequencies to record high resolution soil moisture measurements and freeze/thaw detection at global scale. Unfortunately, shortly after the launch a hardware failure caused the radar to stop working, leaving the radiometer as the only operational mechanism to record data [23]. Since the launch, the radiometer aboard the observatory has been collecting data at a spatial resolution of 36 km and providing global coverage every 2 to 3 days [23]. Observations from SMAP will provide improved estimates of water, energy, and transfers between land and atmosphere [24,25]. SMAP uses lower frequency microwave radiometry (L Band) to map soil moisture at Earth's land surface because at lower frequencies the atmosphere is less opaque, vegetation is more transparent, and the results were more representative of the soil below the skin surface than when higher frequencies were used [26,27]. This research utilizes the SMAP Level 2 enhanced passive soil moisture product (L2_SM_P_E), which is available on a 9-km grid for downscaling to 1-km resolution.

### 2.2. GLDAS Data

NASA's global land data assimilation system (GLDAS) was designed to combine satellite- and observation-based data to produce high resolution, global information on Earth's land surface states and fluxes [28]. GLDAS is able to provide 36 land surface fields from 2000 to the present, including soil moisture, surface temperature, surface runoff, and rainfall. The product of 3-hourly data (GLDAS_NOAH025_3H) with $0.25° \times 0.25°$ spatial resolution was used in this study [29].

Our downscaling approach utilized soil moisture with 0 to 10 cm depth and surface skin temperature from GLDAS that corresponded to the closest overpass times of the Aqua satellite for the LMB, which was approximately 12:00 and 24:00 local time.

### 2.3. MODIS Data

NASA's moderate resolution imaging spectroradiometer (MODIS) was launched aboard the Earth observing system (EOS) aqua satellite in May 2002 and provides atmospheric, terrestrial and oceanic data products [30]. With 36 spectral bands, the highest of any global coverage moderate resolution imager, and spatial resolution ranging from 250 m to 1 km, MODIS is able to provide a multitude of global land products [30]. In this study, daily normalized difference vegetation index (NDVI) and land surface temperature (LST) from MODIS were used to downscale SMAP soil moisture estimates. The 1 km daily LST (MYD11A1), 1 km biweekly NDVI (MYD13A2), and 500 m biweekly climate modeling grid (CMG) NDVI (MYD13C1) were utilized in this study.

### 2.4. AVHRR Data

The advanced very high resolution radiometer (AVHRR) utilizes National Oceanic and Atmospheric Administration (NOAA) polar-orbiting satellites to provide four- to six- band multispectral global data [31]. The AVHRR is used to remotely detect cloud cover and the Earth's surface temperature (NOAA satellite information system, 2013). Prior to MODIS data, AVHRR's 5 km CMG NDVI data were used for long-term surface ground measurements [11]. In this study, daily NDVI data (AVH13C1) from AVHRR from 1981 to 1999 were used. The quality of AVHRR data after this time period is inadequate due to satellite drifting and, therefore, data after 2000 was not used in this study [11] (Table 1).

**Table 1.** Description of the data products used in the downscaling process including their spatial and temporal resolutions and data availability.

| Data Product | Variable | Spatial Resolution | Temporal Resolution | Availability |
|---|---|---|---|---|
| SMAP | Soil moisture | 9 km | Daily | 2015–present |
| GPM IMERG | Precipitation | 10 km | Daily | 2000–present |
| GLDAS | Soil moisture | 25 km | 3 hours | 1979–present |
| MODIS | Land surface temperature (LST) | 1 km | Daily | 2002–present |
| MODIS | NDVI | 1 km | Biweekly | 2002–present |
| AVHRR | NDVI | 5 km | Daily | 1981–1999 |

## 3. Methodology

In this research, the downscaling algorithm and methodology used were developed in Fang et al., 2018 [32]. Similar to the study by Lakshmi and Fang (2015) of the Little Washita Watershed, this study assumes that LST is a linear combination of soil and vegetation temperature [33]. We assume the top soil moisture layer is a function of soil evaporation efficiency and field capacity. It is assumed that the soil moisture at a certain time during the day is inversely proportional to the daily temperature change for the same day, and that the presence of vegetation (NDVI) will influence the soil moisture–temperature change relationship. We also assume that the thermal inertia relationship between temperature difference and soil moisture within a 25 km domain has no spatial variability. Additionally, the assumption is made that the field capacity of each NLDAS pixel is homogenous and does not account for variation at the 1 km scale [32].

In this study, we applied an algorithm developed by Fang et al., 2018, based on soil moisture, LST, and NDVI, to create 1 km soil moisture maps [32]. The methodology of this algorithm is outlined in Figure 2. Due to the effects of vegetation cover on soil moisture estimation, the algorithm applied

here uses a vegetation-based lookup table to relate microwave polarization to soil moisture estimates. As soil becomes more wet its heat capacity increases. The soil moisture at a given time is inversely proportional to the change in temperature 12 hours beforehand, which corresponds with SMAP AM and PM overpasses. Soil moisture daily values were negatively related to the daily temperature difference under varying vegetation conditions. The following equation represents the linear relationship between soil moisture and temperature difference for a specific NDVI (single month):

$$\theta(i,j) = a_0 + a_1 \Delta T_s(i,j) \tag{1}$$

where $\theta(i,j)$ is GLDAS soil moisture gridded to match SMAP overpasses and $\Delta T_s(i,j)$ is the GLDAS 12 h temperature difference closest and prior to SMAP overpasses. This equation uses data at the GLDAS spatial resolution for soil moisture and surface temperature for single months, beginning in 1981. Using the nearest neighbor method, daily NDVI from AVHRR was aggregated to corresponding GLDAS pixels. The NDVI data were categorized into classes from 0 to 1 with increments at 0.1. Classes with less than 8 data points were not included because a sample size smaller than this will not yield valid and statistically significant results from linear regression fitting. Soil moisture at 1 km resolution was calculated from 1 km MODIS LST difference at the corresponding NDVI class. We applied the linear regression fit equation between $\theta$ and $\Delta T_s$, which was built at 25 km resolution, to all the 1 km MODIS grids within the 25 km GLDAS grid. We assumed that the thermal inertia relationship between temperature difference and soil moisture within the 25 km domain had no spatial variability. The following equation represents the correction of the 1 km soil moisture pixel from the MODIS LST products, acquired by removing the difference between SMAP and MODIS derived soil moisture:

$$\theta^{corr}(i,j) = \theta(i,j) + \left[ \Theta - \frac{1}{n} \sum_{i=1}^{n} \Theta_i \right] \tag{2}$$

where $\theta^{corr}(i,j)$ is the corrected 1 km soil moisture, n is the number of 1 km soil moisture pixels that are in each SMAP 9 km pixel, $\Theta$ is the original SMAP 9 km soil moisture estimate, and $\theta_i$ is the number of uncorrected 1 km SMAP soil moisture pixels that fall in the original 9 km SMAP grid $\Theta$. The value of n is ideally 81, but it may be less due to cloud contaminated data. The corrected soil moisture was characterized by the soil moisture and daily temperature relationship, which changed under different vegetation conditions. Since visualizing rainfall is essential to determining the response of the soil moisture, rainfall from GPM IMERG was used in this study to analyze the wetting and dry-down patterns after a significant rainfall event. One limitation of this methodology occurred when the 9 km original SMAP was biased. Then that bias was passed onto the corrected 1 km soil moisture. Another limitation was the difficulty to recover cloud-contaminated data, which resulted in spatial inconsistencies in the 1 km corrected soil moisture maps.

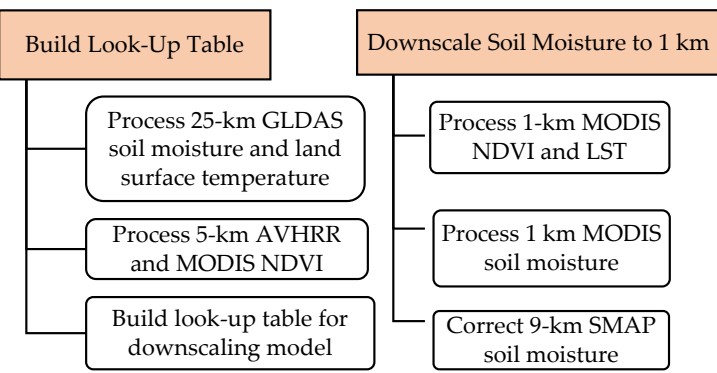

**Figure 2.** Workflow for building downscaling model and executing the algorithm.

The algorithm used in this study was validated using in situ measurements in the CONUS region, by Fang et al., 2018, for soil moisture estimates from AMSR2 between 2015 and 2017. Their validation showed variability in seasonal performance and stronger correlations in the soil moisture–temperature change relationship during summer months. Also, the remotely sensed soil moisture and downscaled estimates both underestimated in situ soil moisture during precipitation events. It is important to note the effects of precipitation on soil moisture retrieval; the microwave sensing depth is reduced. An additional validation of this algorithm was performed in the Walnut Gulch Experimental Watershed (WGEW) and indicated that downscaled soil moisture had better validation metrics than the original SMAP [32]. The $R^2$ of the 1 km soil moisture ranged from 0.189 to 0.697, whereas the 9 km SMAP ranged from 0.003 to 0.597. The slope values for the 1 km are higher than those for the 9 km SMAP. Additionally, the 1 km soil moisture RMSE values and biases improved compared to the original SMAP data. There were no consistent soil moisture measurements in the Lower Mekong Basin, and this presents a formidable challenge to validation. However, future work may be able to carry out validation by comparison of the 1 km soil moisture to outputs from hydrological models.

## 4. Results

### 4.1. Rainfall Variation in the Lower Mekong Basin

Variations in rainfall patterns result in changes in soil moisture. Precipitation has a direct impact on the wetting and drying of soils and, therefore, must be examined alongside soil moisture. In the LMB, the annual wet season (April–September) results in more vegetation growth and cloud cover compared to the dry season. Therefore, the ability to measure soil moisture via remote sensing is affected during these months. Daily precipitation data from GPM IMERG Final Precipitation L3 1 day 0.1° by 0.1° V05 (GPM_3IMERGDF) were aggregated for monthly accumulation for April through September from 2015 to 2018, to correspond with the downscaled soil moisture in order to examine the monthly variations (Figure 3) [34].

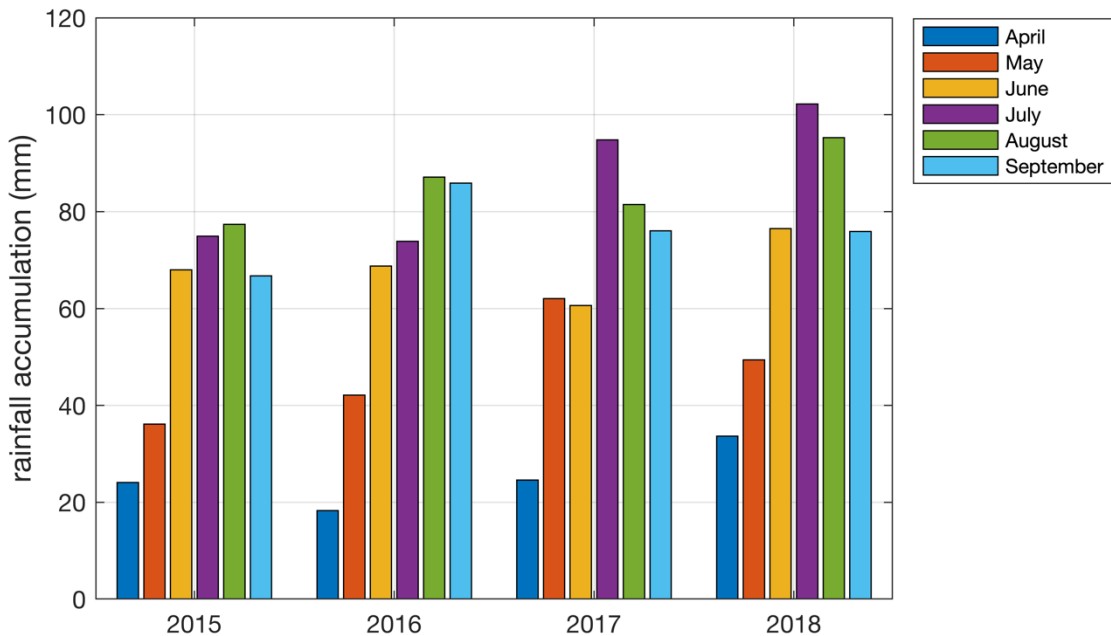

**Figure 3.** Bar plot of monthly average precipitation for April 2015–September 2018 in the LMB.

In this study, precipitation patterns varied in the wet season months, with July and August generally accumulating the most rainfall and April and May receiving the least (Figure 3). Additionally, precipitation varied from year to year over the LMB, with certain years being more dry or wet than others due to regulation by monsoons. For example, comparing the year 2016 to 2018 in Figure 3

shows 2016 as a much dryer year, especially in the wettest month of the year, July, which received over 100 mm of rainfall. This pattern can also be seen by comparing the monthly maps from 2016 and 2018 (Figure 4). Figure 4 shows the spatial distribution of accumulated precipitation over the LMB for each month, corresponding to the 1 km soil moisture estimates. Rainfall patterns varied significantly between countries in the LMB. Areas in Laos and Cambodia receive the greatest amounts of precipitation annually (over 2800 mm), while the Thailand plateau only received a third or less of that amount. Here, precipitation from IMERG was used to detect and observe the dry-down patterns of soil moisture after a large rainfall event.

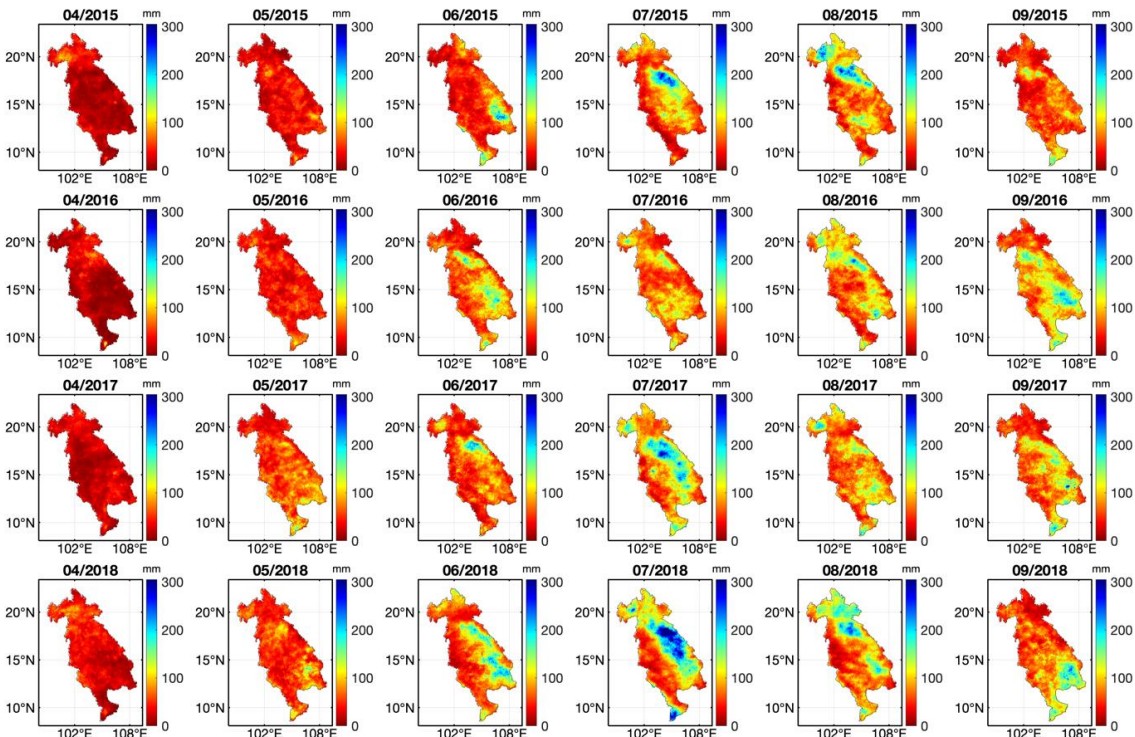

**Figure 4.** Monthly rainfall accumulation from GPM IMERG for April 2015–September 2018 in the LMB.

### 4.2. Soil Dryness Response to Large Rainfall Events

In this section, soil moisture is examined alongside precipitation with the purpose of examining the drying of soil over time in response to a rainfall event. By evaluating the time series after a large precipitation event with almost no subsequent precipitation, we were able to observe the near-surface soil moisture observations as they transitioned from saturated to dry conditions. Daily 9 km SMAP soil moisture estimates were compared to daily 10 km IMERG precipitation to examine the response of soil moisture to precipitation events. It is possible that, in the absence of precipitation, agriculture is irrigated. Hence, we may have seen wetness from irrigation in these regions, despite no significant rainfall event. Figure 5 shows the relationship between daily rainfall and soil moisture between 2015 and 2018 averaged over the LMB.

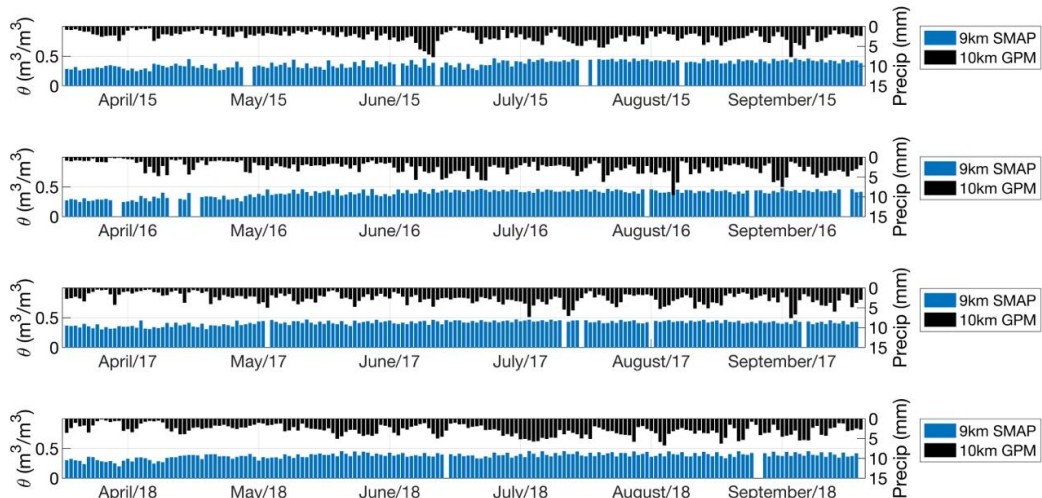

**Figure 5.** Time series of daily soil moisture active passive (SMAP) 9 km soil moisture and daily Global Precipitation Measurement-Integrated Multi-satellitE Retrieval (GPM-IMERG) 10 km precipitation for April 2015–September 2018 averaged in the LMB.

Using Figure 5, two precipitation events were selected in which soil moisture exhibited a clear dry-down pattern after the rainfall. The events were examined more closely in combination with corresponding daily downscaled soil moisture, in order to evaluate the improvement in the representation of drying from 9 km to 1 km. Figure 6 more closely examines the time series of the dry-down period in the LMB from 13 April 2015 to 20 April 2015, after a large precipitation event occurred on 13 April. Figure 7 shows the spatial distribution of rainfall, 9 km SMAP soil moisture, and 1 km downscaled soil moisture for each day during the dry-down period. The second event selected was from 6 April 2018 to 11 April 2018. Figure 8 shows the time series of the dry-down period after the precipitation event on 6 April 2018. The 1 km soil moisture (blue) was better able to capture the dry-down pattern than the 9 km SMAP soil moisture (green) (Figure 8). Figure 9 shows the spatial distribution of rainfall, 9 km SMAP soil moisture, and 1 km downscaled soil moisture for each day during the dry-down period in April 2018. The coverage of the 1 km corrected soil moisture was dependent on MODIS LST data and influenced by cloud cover, which made it difficult to find good coverage on consecutive days. The 1 km SMAP did not perform as well during wet days due to the spatial coverage of the MODIS land surface temperature (LST) data being compromised by cloud contamination.

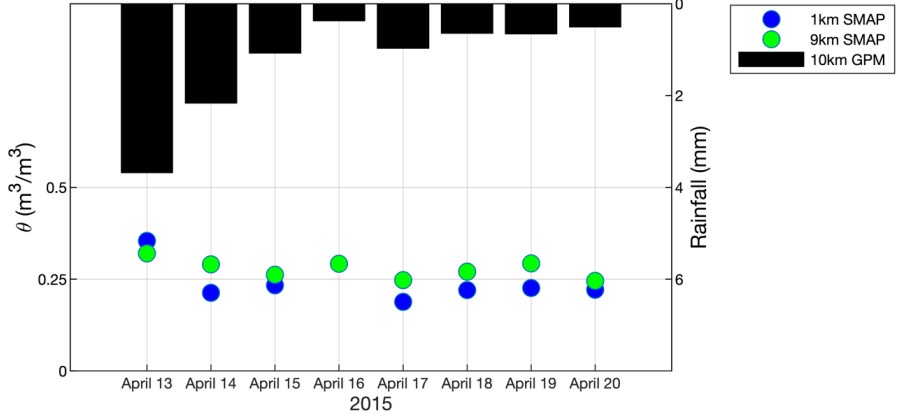

**Figure 6.** Time series of dry-down event from 13 April 2015 to 20 April 2015 with 9 km SMAP soil moisture (green), 1 km downscaled soil moisture (blue), and 10 km rainfall data from GPM IMERG (black).

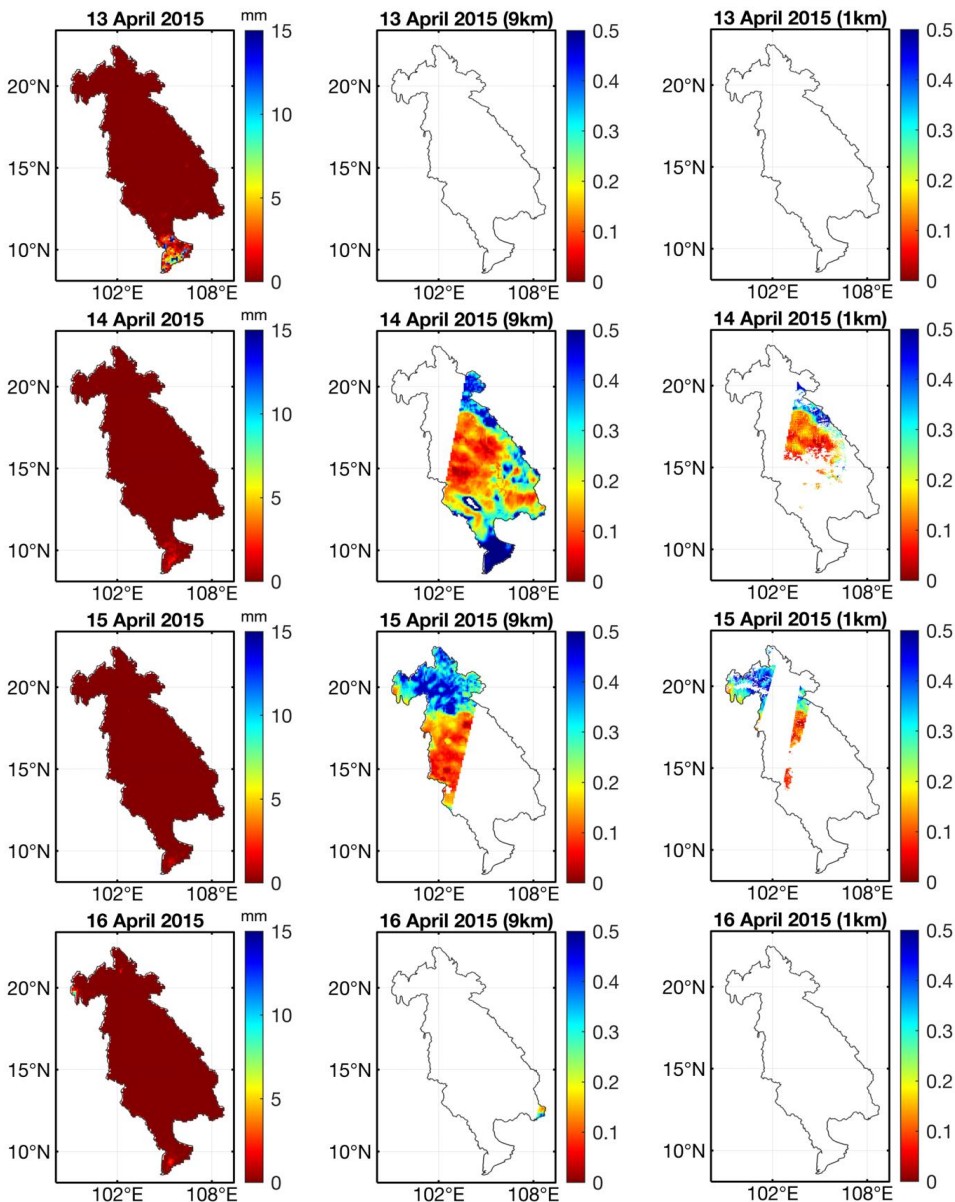

**Figure 7.** *Cont.*

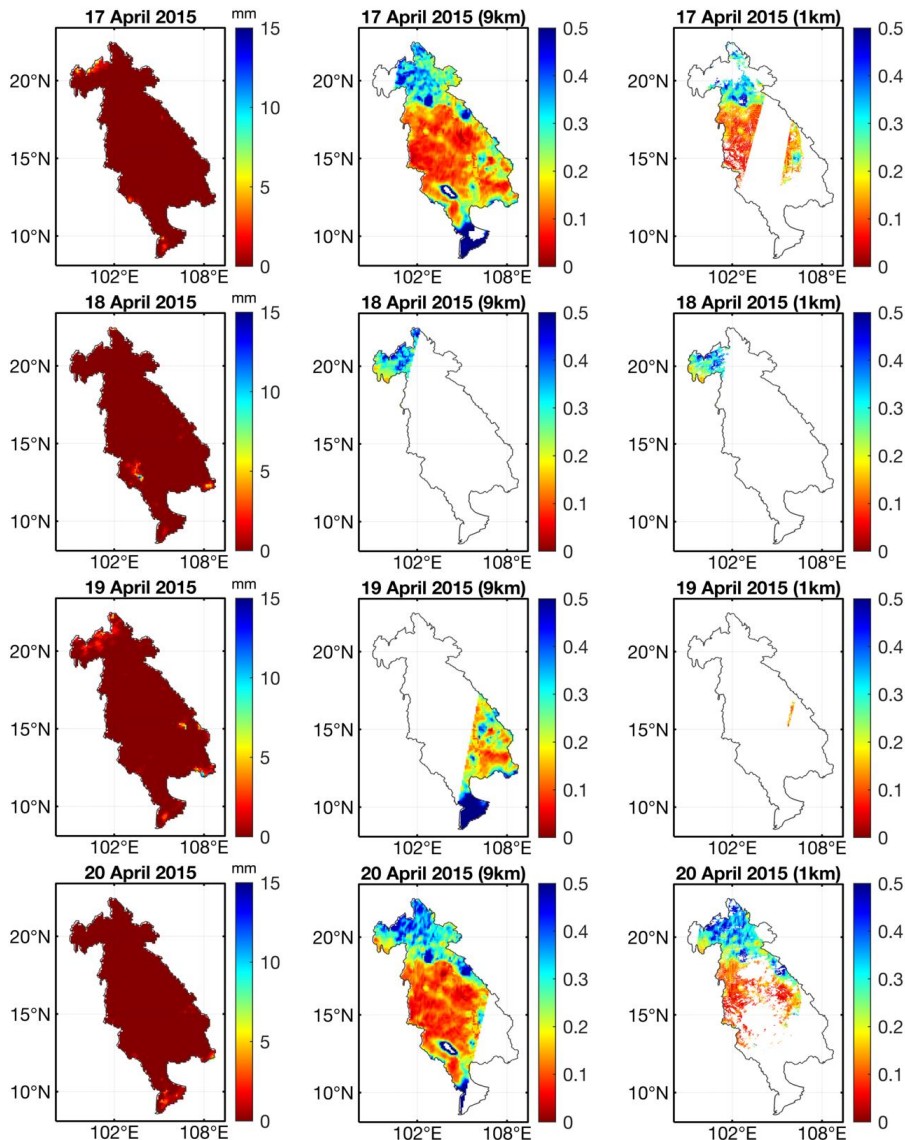

**Figure 7.** Dry-down event for 13 April 2015 to 20 April 2018 represented by 10 km IMERG rainfall, 9 km SMAP, and 1 km downscaled soil moisture.

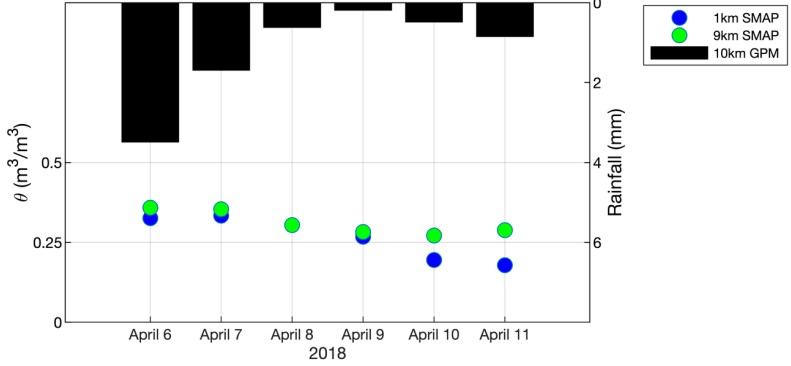

**Figure 8.** Time series of dry-down event from 6 April 2018 to 11 April 2018 with 9 km SMAP soil moisture (green), 1 km downscaled soil moisture (blue), and 10 km rainfall data from GPM IMERG (black).

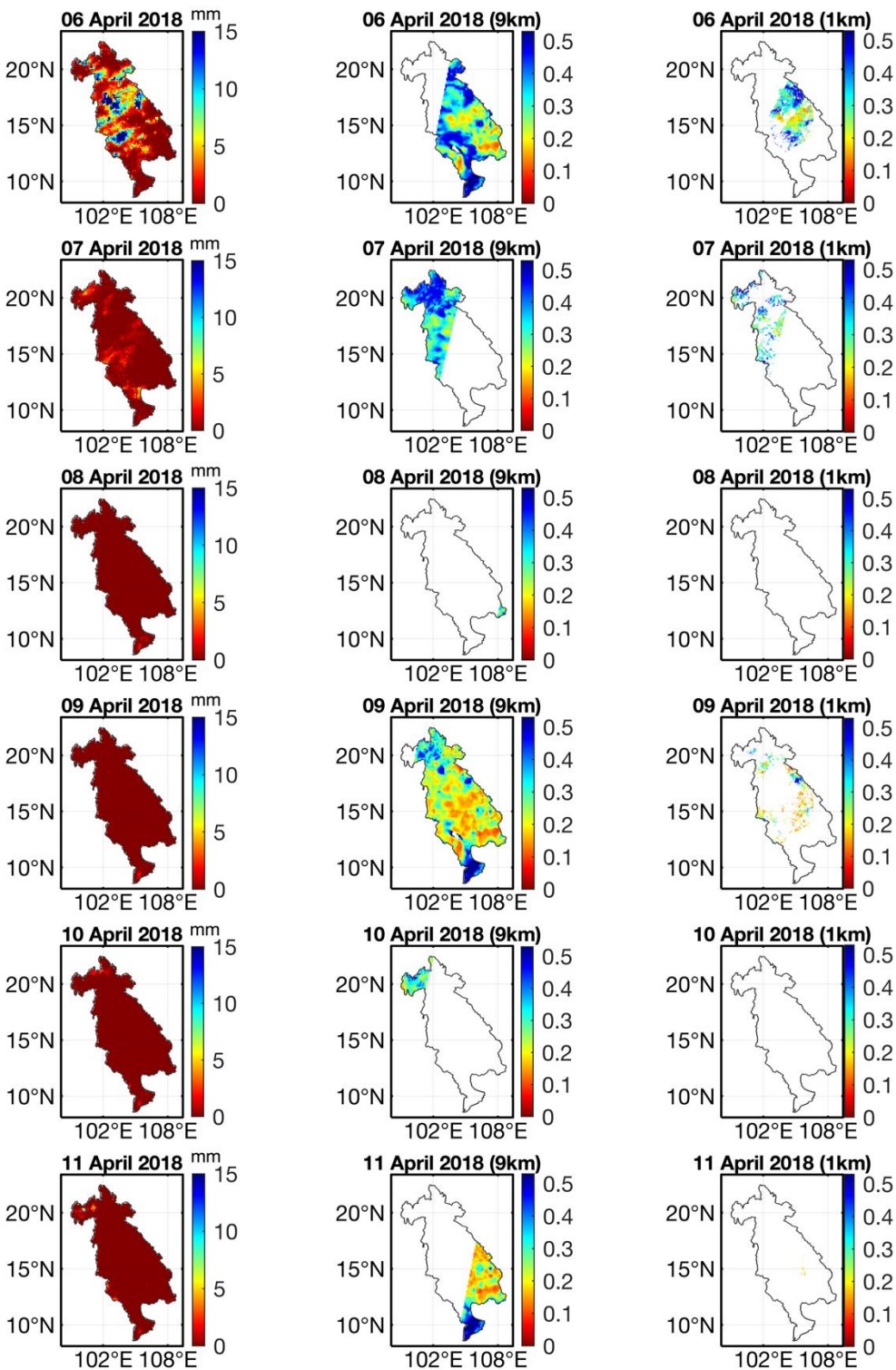

**Figure 9.** Dry-down event for 6 April 2018 to 11 April 2018 represented by 10 km IMERG rainfall, 9 km SMAP, and 1 km downscaled soil moisture.

*4.3. Importance of High Spatial Resolution Soil Moisture for Hydrology and Water Resources*

The high spatial resolution observed soil moisture generated in this study was an important data set that could not be obtained from other sources. Firstly, there are no consistent in situ networks that monitor soil moisture in the Lower Mekong River Basin. Even in other parts of the world that do have

such networks, they are seldom dense enough to produce soil moisture at 1 km spatial resolution. Secondly, although land surface models can simulate soil moisture at high spatial resolution, they lack the precipitation input at 1 km spatial resolution, which is needed to minimize variations in small-scale processes [35]. Currently, the "best" spatial resolution of globally available precipitation is the climate hazards group infrared precipitation with station observations (CHIRPS) at 0.05°. CHIRPS provides estimates from 1981 to the near present and uses a recently produced satellite rainfall algorithm that combines climatology data, satellite precipitation estimates, and in situ rain gauge measurements to produce a high resolution precipitation product [36].

The 1 km spatial resolution soil moisture from this research can be used in combination with land use and land cover data from MODIS (moderate resolution imaging spectroradiometer) at 1 km and Landsat imagery at 30 m to map the co-variability of land use and wetness. This will be a valuable tool for land use planning, specifically in the LMB where there are competing cropping strategies and land use for industrial development. Additionally, this 1 km soil moisture can be used to determine antecedent soil moisture conditions in watershed modeling, meaning it can serve as an input to determine the portion of rainfall that will infiltrate the soil and that which will run off to the stream network. More detailed estimations of streamflow runoff will in turn benefit flood prediction and monitoring in watersheds [37]. This high spatial resolution 1 km observed soil moisture can serve a variety of water resource applications and will be of much use in the LMB.

## 5. Conclusions

This study applied a previously developed method to a new geographical location where in situ observations are lacking. Here, higher resolution could help various land use decisions such as construction of dams, agriculture, and aquaculture. In this study, soil moisture estimates of the Lower Mekong River Basin from April 2015–September 2018, from SMAP Enhanced L2 Radiometer Half-Orbit 9 km V.2., were enhanced to 1 km resolution. In this study, we applied an algorithm developed by Fang et al., 2018, based on soil moisture, LST, and NDVI to create 1 km soil moisture maps. Soil moisture daily values were negatively related to the daily temperature difference under varying vegetation conditions. The downscaling algorithm was based on LST, soil moisture, and NDVI and used the relationship between daily soil moisture and daily land surface temperature difference between satellite overpasses as well as the vegetation class to downscale soil moisture to a higher resolution. The months of April and May showed the best coverage of soil moisture at 1 km and July–September showed the least coverage at 1 km, due to LST/NDVI data with substantial cloud coverage and higher vegetation growth. It was discovered in this study that the 1 km SMAP did not perform as well during wet days due to the spatial coverage of the MODIS land surface temperature (LST) data being compromised by cloud contamination.

Soil moisture estimates are readily available at global scale from a multitude of satellite products but are represented at spatial scales that are often too coarse for effective hydrological modeling and decision-making purposes. Soil moisture at high resolution can be used in place of ground measurements for land and water management decisions in large river basins where in situ data are limited such as the LMB. The high resolution soil moisture estimates derived in this study can be more useful for assessing dry-down and wetting trends than coarser resolution data, such as the 9 km SMAP product in the LMB. Additionally, 1 km soil moisture retrievals can better aid drought and crop productivity monitoring, flood forecasting, and assist weather forecasting by providing greater spatial representation than coarser products. This high spatial resolution soil moisture at 1 km can be applied to a multitude of water resources applications in order to benefit large watershed management.

**Author Contributions:** Individual contributions from the authors include conceptualization, C.D.; B.F.; V.L.; methodology, B.F.; software, C.D.; validation, B.F.; V.L.; formal analysis, C.D.; B.F.; investigation, C.D.; resources, V.L.; data curation, C.D.; and B.F.; writing—original draft preparation, C.D.; writing—review and editing, C.D.; B.F.; V.L.; visualization C.D.; B.F.; V.L.; supervision, V.L.; project administration, V.L. All authors have read and agreed to the published version of the manuscript.

**Funding:** This research received no external funding.

**Conflicts of Interest:** The authors declare no conflict of interest.

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
