# Peer review of "Downscaling of SMAP Soil Moisture in the Lower Mekong River Basin"

_water, doi:10.3390/w12010056_

Round 1

Reviewer 1 Report

This manuscript presents an application of a downscaling method previously developed by the same coauthor to the generation of 1-km SMAP soil moisture in the Lower Mekong River Basin (LMB).  This application highlights the importance of using satellite-based data in areas where ground measurements are rare or even inexistent, so the manuscript should merit publication in Water.  Nevertheless, there is some notable insufficiency of this study that the authors can enhance before publication.  Specific comments are provided below:

Starting from the title, I can't find any realistic analysis associated with using downscaled soil moisture data "in support of water management decisions" in the manuscript.  There is only a short section (Sec. 4.3) showing some quite general, high-level discussion regarding the implication of having high-resolution soil moisture data for hydrology and water resources.  You should either revise the title or include some realistic analysis (e.g., using 1-km soil moisture in hydrological modeling) in the manuscript.  One can easily criticize this manuscript being short of independent contribution to the filed of remote sensing or water resources in general because the key method is not a new invention.  Changing the article style to case study would solve this issue, or the authors are required to stress those new concepts/findings (if any) other than simply applying an existing method to a different region. Probably the original methodology paper (i.e., Fang et al., 2018) has shown the validation of the downscaling method using ground measurements; however, the validation results might be (in my opinion, should be) region-dependent.  There seems to be no ground measurements available in the LMB, which poses another notable issue that the downscaled soil moisture product may be unreliable.   As the matter of fact, the key question to ask is that do we really want to have the 1-km product under the condition that the original data at a coarser resolution already exhibit substantial biases?  Look at the 10-km GPM precipitation data, do we really need 1-km soil moisture?  If you still believe so, could you quantify the potential biases and uncertainty associated with the 1-km product? While you use the established method, please be more specific and cautious when summarizing the method; all assumptions and limitations should be clearly addressed as well.  Equation 1 is derived using data (GLDAS) at 25-km resolution, but later the same equation is used to estimate soil moisture at 1-km resolution; does that mean the same regression coefficients (a0 and a1) within a 25-km pixel are repeatedly used to generate data at 1-km resolution (i.e., 25x25 repetitions)?  Why do you exclude NDVI classes with less than "8" data points?  There should be no subscript "n" for θ in the summation; is n less or equal to 81 (9x9)? There is another assumption that the difference between SMAP and GLDAS at 9-km resolution carries over to 1-km resolution; that is, each 1-km pixel within a 9-km pixel is corrected using the same difference value obtained from the 9-km relationship (which could be problematic).  Last but not least, from Figures 7 and 9, clearly the downscaled product does not bear the same spatial extent with the original one; is it due to the lack of 1-km LST or what? Abstract at its current form is too lengthy; some background information (first few sentences) could be removed or shortened. One more column of "variable" should be added to Table 1. Figure 2 does not look like a formal workflow/flowchart.  For instance, "Process data" is not comparable to other boxes, and most boxes are actually data rather than process.   The use of GPM does not add much value to this work.  Figures 6 and 8 that show the time series of GPM and SMAP at a random pixel are not so representative.  A more systematic approach that covers more spatial information (more pixels) should be performed.  

Reviewer 2 Report

This is an interesting article. It is well written and well organized. Looks like the 9km SMAP fits well with 1 km SMAP. I have one question regarding Figure 5. Why even in the dry condition the soil moisture is still above 25-30%? Is it because of irrigation? For example in April'15 to June'15.

Round 2

Reviewer 1 Report

The authors have addressed most of my comments.  However, I notice that the current article type is still "article" instead of "case study;" especially the authors have admitted that this study simply applied a previously developed method to another region.  Some other follow-up comments: 1) Since you have acknowledged that the 1-km results cannot be validated in the LMB, and are subject to cloud contamination, you should make it very clear that the 1-km soil moisture only represent an surrogate product, not necessarily an improved one.  2) In my previous report, the last two sentences actually constitute one same comment; and according to your response, it seems that the relationship between GPM and SMAP is hard to obtain during wet days (due to cloud cover again).  Does this imply the 1-km SMAP may not perform well during wet days (at least the spatial coverage has been compromised)?
